# Carbon Loaded Nano-Designed Spherically High Symmetric Lithium Iron Orthosilicate Cathode Materials for Lithium Secondary Batteries

**DOI:** 10.3390/polym11101703

**Published:** 2019-10-17

**Authors:** Diwakar Karuppiah, Rajkumar Palanisamy, Subadevi Rengapillai, Wei-Ren Liu, Chia-Hung Huang, Sivakumar Marimuthu

**Affiliations:** 1#120 Energy material lab, Department of Physics, Alagappa University, Karaikudi 630003, Tamil Nadu, India; selfindicator@gmail.com (D.K.); rajphysics@yahoo.com (R.P.); 2Department of Chemical Engineering, R&D Center for Membrane Technology, Research Center for Circular Economy, Chung-Yuan Christian University, Chung-Li 32023, Taiwan; wrliu@cycu.edu.tw; 3Metal Industries Research and Development Centre, Kaohsiung 81160, Taiwan; chiahung@mail.mirdc.org.tw

**Keywords:** Li_2_FeSiO_4_, cathode, lithium batteries, polyol method

## Abstract

In the present study, Li_2_FeSiO_4_ (LFS) cathode material has been prepared via a modified polyol method. The stabilizing nature of polyol solvent was greatly influenced to reduce the particle size (~50 nm) and for coating the carbon on the surface of the as-mentioned materials (~10 nm). As-prepared nano-sized Li_2_FeSiO_4_ material deliver initial discharge capacity of 186 mAh·g^−1^ at 1C with the coulombic efficiency of 99% and sustain up to 100 cycles with only 7 mAh·g^−1^ is the difference of discharge capacity from its 1st cycle to 100th cycle. The rate performance illustrates the discharge capacity 280 mAh·g^−1^ for lower C-rate (C/20) and 95 mAh·g^−1^ for higher C-rate (2C).

## 1. Introduction

An economic status of any nation has been scaled by consumption of electric energy. All the countries are probing viable technology for electric energy storage devices from the intermittent renewable sources. Complacency of the instantaneous demand can be fulfilled with the help of a storage system. Solid state batteries have stored the electric energy in the form of a chemical reaction. Usually, batteries comprises of the anode, cathode, separator and electrolyte [1,2,3,4,5,6]. This storage system was highly recommended for replacing the combustion engines, because of its properties like more silent, no CO_2_ emission, cost effective [7,8,9,10]. Battery technologies can be classified according to their intercalated alkaline metals such as lithium [11,12,13,14,15], sodium [16], potassium [17], magnesium [18], etc. Among these, lithium based technology is one of the promising candidate owing to its increased volumetric and power density [19,20]. Cost and less lithium resources is a major drawback for implementing the Li-ion storage system [21,22]. Hence, a special attention has been focused with respect to the battery components. In batteries, the cathode ascertained capacity of the whole system (i.e., higher amount of Li^+^, leads to boost up the capacity) [23]. The anode allows reversible insertion/extraction of lithium ions, which was released from the cathode with the help of electrolyte (contain ions) through the separator [24,25]. Among all the components, the cathode holds the predominant place to judge the performance of the battery. Commercially used cathode material includes layered LiCoO_2_ [26,27] exhibits higher voltage but more toxicity, spinel LiMn_2_O_4_ [28] provides excellent rate capability with poor cycling. Olivine LiFePO_4_ [29,30] has good structural stability during cycling due to the presence of polyanion and the only demerit is poor electronic conductivity. However, its theoretical capacity is only 170 mAh·g^−1^. In order to achieve twice the theoretical capacity, polyanionic Li_2_FeSiO_4_ (LFS) was introduced by Nyten et al. [31], because of the charge of silicate anionic is higher than that of phosphate. However, all other parameters are similar to phosphate. The orthosilicate exhibits better thermal stability, because they possess a framework that comprises of strong covalently bonded Si–O, which reduces the risk of ignition during cycling. However, the major demerit of the LFS cathode suffered by the poor conductivity of order 6 × 10^−14^ S·cm^−1^, leads to control the utilization of its full theoretical capacity [32]. The materials synthesized via solution based routes, low temperature method offers nano-sized powder, creating more channels for the lithium diffusion path way, because Li^+^ diffusion plays a significant role in the performance of the cathode, especially for commercial application [33]. The coating of higher conducting materials like conductive carbon, metal oxide, etc. was coated on the surface of the nano sized LFS. To date a lot of different synthesis approaches have been followed to prepare phase pure Li_2_FeSiO_4_ in most applied solution based techniques including sol-gel [34], hydrothermal [33], polyol [35], spray technology [36], super critical fluid [37], solvo-thermal [38], the co-precipitation method [39], etc. Among these most cited hydrothermal methods needs autoclave during the synthesis process, this is maintained at a particular pressure. Hence it is so difficult to adopt this method for large-scale production. However, in the case of the polyol method, it offers an ambient synthesis condition i.e., atmospheric pressure and higher degree of crystallites with homogeneous distribution of particle. The cathode materials Li_2_FeSiO_4_ (LFS) capture a wide attention due to its high theoretical capacity (of 332 for 2 Li^+^ and 166 mAh·g^−1^ for 1 Li^+^ diffusion), safety and environmental benignity [40]. However, it has been suffered from poor practical capacity attributed to the poor electronic conductivity and presence impurities, which were generated during the synthesis process [32,41]. In order to control the impurities, each and every step of synthesis process must be concerned. Initially, the solvents such as water, ethanol and methanol used to hydrolysis the tetra ethoxysilane, which is the major starting material for silicate. During condensation the silicon oxide particles aggregate together by using the as-mentioned solvent, but the glycol-based solvent minimizes the aggregation of particles. Due to the stronger polarity nature of ethylene glycol, it makes a weaker hydrophobic effect. Thus fast hydrolysis kinetics and well homogeneous morphology has been obtained. According to stokes Einstein relation $D=\frac{1}{\eta }$ (where *D*: diffusion constant and *η*: viscosity of the solution) the much higher viscous of ethylene glycol 21 mPa s, 20 °C as compared to water 1.0087 × 10^−3^ mPa s, 20 °C decrease the diffusivity of metal ions in the solution like Fe^2+^ and Li^+^, which regulate the direction of crystal orientation and growth [42,43]. The different types of polyalcohol solvents include ethylene glycol, di, tri, tetra and poly ethylene glycol. While using tetra and ethylene glycol as a solvent the particle obtained with nano sized towards inhomogeneous particle distribution. Hence, diethylene glycol is providing homogeneous particle. Extraordinary architecture electrode materials with spherical morphology in the form of nanosized particle with carbon composite could be easily yielded by using the polyol method and so on.

In this regard the low temperature method has been adopted to obtain nanoparticle via polyol. The major advantage of polyol over hydrothermal is that polyol works in atmospheric pressure. Particularly, the low temperature method facilitates controlled morphology compared with the high temperature method. This work discloses the preparation and the half-cell performance of spherically symmetric Li_2_FeSiO_4_/C particle using the polyol method, since it is an energy efficient route. This method offers worthy homogeneity and reduced particle size. The structural, elemental, morphological and electrochemical performances of the Li_2_FeSiO_4_/C materials have been characterized by XRD, XPS, SEM and galvanostatic charge–discharge analyses. 

## 2. Materials and Methods

LFS/C was prepared by the polyol method using diethylene glycol (DEG) (99%, Alfa Aesar) as a solvent [44]. Stoichiometric amounts of iron (II) acetate (Alfa Aesar), tetraethyl orthosilicate (99%, Alfa Aesar) and lithium acetate (99% Alfa Aesar) were mixed and dissolved using DEG and stirred by adding few drops of H_2_SO_4_ to catalyze the reaction. The solution was reflexed overnight at 245 °C to obtain nano-sized LFS/C particles. This solution was centrifuged to separate the precipitant. The obtained precipitate was dried at 150 °C to get rid of the solvent. Then the dried resultant powder was sintered at 600 °C for 8 h in Ar atmosphere and was quickly transferred to the glove box to avoid oxidation.

An X-ray diffraction (XRD) measurement was performed to investigate the crystallographic information using powder X-Ray Diffractometer (X’Pert Pro-PAnalytical, The Netherlands, 2007) with monochromated Cu–K-Alpha X-ray source. The elemental analysis was analyzed with the help of X-ray Photoelectron Spectroscopy (XPS) (PHI-VERSAPROBE III, Japan, 2018). The vibration spectrum was analyzed by Fourier Transform Infrared (FT-IR) spectrophotometer (Thermo Nicolet 380, USA, 2008). The formation of carbon was identified Raman Spectrometer (SEKI Corporation, Japan, 2011). Morphological images were recorded using High-Resolution Transmission Electron Microscopy (HR-TEM) (JEOL-2100+, Japan, 2018) and Field Emission Scanning Electron Microscope (FE-SEM) (Quanta FEG 250, USA, 2012). Galvanostatic charge/discharge measurement was performed in the potential range from 1.5 to 4.5 V vs. Li/Li+ with a Bio-Logic battery cycling station BCS 815 with an EIS system (Bio-Logic, France, 2018).

The coin cells (2032 type) were assembled inside an argon-filled dry-box with lithium metal as the anode, Celgard 2400 as the separator and 1 mol L^−1^ LiPF_6_ in EC:DMC:EMC (1:1:1 by volume) as the electrolyte. The cathode slurry was made by mixing active materials, carbon black and poly(vinylidene fluoride) binder in a weight ratio of 80:10:10 in the solvent *N*-methyl-2-pyrrolidone (NMP) and the loading mass of the electrode was 3.2 mg·cm^−2^. Then the slurry was coated uniformly on aluminum foil, and then dried at 120 °C for at least 6 h in vacuum. Finally, the as-prepared cathodes were used to assemble cell couple to perform C/D and EIS analyses. 

## 3. Results

The crystalline structural information was studied using XRD analysis and the diffraction patterns of the samples are shown in Figure 1a. 

All the diffraction peaks could be agreed to an orthorhombic Li_2_FeSiO_4_/C with the space group Pmn2_1_ and the lattice parameters were a = 0.628 nm, b = 0.533 nm and c = 0.497 nm and β = 98.92° [45]. Well-defined sharp intense diffraction peaks indicate high degree of crystalline nature. Figure 1b shows the FTIR spectrum of Li_2_FeSiO_4_/C absorption peak around 900 cm^−1^, indicating the stretching vibration of the [SiO_4_]^4−^ polyanion group. Two peaks around 586 and 530 cm^−1^ were identified as the bending vibration of the same polyanion. It is necessary to check the presence of impurity because of the Li_2_SiO_3_ was easily nucleate during the synthesis process [46]. A peak at 780 cm^−1^ was found to be absent, which dictates the absence of impurity Li_2_SiO_3_ and it affirms that the as-prepared LFS/C had a pure phase. Besides, the peak appeared at 815 cm^−1^ was assigned due to the presence of SiO_4_, well matched with the reported one [47]. Raman spectroscopy was performed to study the chemical structure LFS/C. Figure 1c shows that there were two peaks that appeared at 1330 and 1590 cm^−1^, which were attributed to D-band sp^3^–type and G-band sp^2^ type respectively. The magnitude of the ratio of intensity of D and G bands gives degree of graphitization in carbon; i.e., lower magnitude possesses higher order of graphitization [48]. For as-prepared LFS possessed a value of $\;\left(\frac{I_{D}}{I_{G}}=0.93\right)$, which gave rise to high electronic conduction between the particles. The enhanced intrinsic electrical conductivity of LFS/C has favored in improving the electrochemical activity [49]. Further confirms with the help of EIS studies. 

XPS has been used to investigate the oxidation state of elements presence on the surface of LFS electrode. The XPS spectrum for the as-prepared sample is shown in Figure 2a–d. The survey spectrum of the LFS electrode cathode materials hold the characteristic peaks of Fe, O, C, Si and Li as shown in Figure 2a. Further, Figure 2b shows the binding energy of O 1s as 531.43 eV, corresponded to the O^2−^ state of oxygen. The Fe 2p state reveals that there were two peaks at 712.42 eV for Fe 2p_1/2_ and 725.93 eV for Fe 2p_3/2_ due to the presence of Fe^2+^, which was well matched with the reported values [50] in Figure 2c. Figure 2d illustrate the binding energy peak at 284.92 eV, which could describe the presence of C 1s, which was in good agreement with the literature [51]. At the same time, Raman analysis provided some preliminary idea of the presence of carbon on LFS [52].

The actual particle size, crystallite phase and presence of carbon on the surface of as-synthesized LFS electrode material, were investigated through HR-TEM and shown in Figure 3. From Figure 3a,b shows the existence of nano particles about 50 nm in size with spherical morphology. Further, it is clearly noticed that about 15 nm of carbon was found over the surface of LFS, it is similar with a previous report [53], which was in favor of the fact that the residual polyol solvent (DEG) underwent pyrolysis during sintering at 600 °C. Hence, the pyrolysis process initiated carbon formation and acted as an additive for surface modification of LFS and also built a conductive carbon network among LFS particles. The network enhanced the electronic conductivity of LFS electrode. It was already confirmed with the help of Raman analysis by using the ratio of I_G_ and I_D_. Subsequently, it was identified from Figure 3c, wherein the two kinds of crystal lattice fringes with lattice spacing of about 0.513 and 0.265 nm were observed, which corresponded to the orthorhombic Li_2_FeSiO_4_ with the plane (101) and (202), this result coincided with the XRD plane. Figure 3d confirms the polycrystalline nature of Li_2_FeSiO_4_, which was obtained via the polyol route [54].

Figure 4a,b illustrates the rate performance of the as-prepared cathode material LFS/C. It delivered the discharge capacity of 280, 262, 207, 154, 95 and 270 mAh·g^−1^ at different C rates of C/20, C/10, C/5, 1C, 2C and C/20 up to 10 cycles of each current density. These values were comparatively higher with the previous report [55]. Even though, LFS electrode was subjected to high current density of 2C, it retained the low C-rate due to structural stability of the material. After several C-rates, the discharge capacity was reached about 270 mAh·g^−1^ at C/20, respectively. The average coulombic efficiency and capacity retention is revealed in Figure 4b. It can be clearly show that about 99% of coulombic efficiency was obtained. In Figure 4b shows the rate capability average capacity retention relative to the highest capacity (initial lithiation) from a lower C-rate (C/20). The capacity retention of the as-prepared electrode was listed as follows 98%, 92%, 71%, 53% and 31% at lower to higher C-rate C/20, C/10, C/5, 1C and 2C respectively. It is noted that the lower particle size increased the diffusion channel of lithium. 

The galvanostatic charge discharge analysis has been performed in the potential range 1.5–4.5V at 1C-rate. Figure 5a,b show that discharge capacity of the LFS at 1st and 100th cycle was 186 and 179 mAh·g^−1^ (with the coulombic efficiency of 99% and sustained even up to 100 cycles with only a difference of 7 mAh g^-1^, initiated from the 1st and 100th cycle) respectively. The capacity retention of 96% was attained for this material, which was comparatively higher than reports for other orthosilicates [54,55]. At the same time, the initial charge peaks rose at higher voltages of approximately 4.3 V, and the discharge peaks raised at lower voltages of about 1.82 V, correspond to the potential plateau in the charge–discharge curves, respectively [43]. It provides better discharge capacity, coulombic efficiency and capacity retention as compared to previous reports [56,57,58,59]. The betterment in the performance of Li_2_FeSiO_4_ was obtained with the uniformly nano architecture symmetry spherical particles with proper loading of carbon. Comparable also with the report for Li_2_FeSiO_4_ synthesized using the polyol method [60]. From EIS, the Nyquist plot of LFS/C (Figure 6a) reveals that the *R_ct_* value of fresh cell and 10th cycle was 7.5 and 19 Ω. Increases in the values was due to the formation of solid electrolyte interface upon cycling [52]. The unwanted side reaction taken place at the electrode and electrode interface in the half cell, led to increase the charge transfer reaction [61]. Furthermore, the lithium ion diffusion coefficient (*D_Li_*_+_) was calculated using the Warburg impedance factor *σ*, which was obtained from the slope of Figure 6b.
(1)$$D_{Li^{+}}=\frac{R^{2}T^{2}}{2A^{2}\;n^{2}\;F^{4}\;C^{2}\;\sigma^{2}}$$
where, *R* is the gas constant (8.314 J·mol^−1^·K^−1^), T is the absolute temperature (298 K), *n* represents the number of electrons transferred, *F* represents Faraday constant (96,485.33 C·mol^−1^), *A* means the surface area of the electrode (0.785 cm^2^) and *C_Li_*_+_ (0.0390 mol·cm^−3^) means the concentration of lithium ions. *σ* is the Warburg factor by using the diffusion coefficient of as prepared LFS was measured as 1.67 × 10^−16^ cm^2^·s^−1^. It is an order of magnitude higher than reported phase pure materials [62,63] that leads to increase the capacity during cycling. 

It is noticed from Figure 7a,b that there not many apparent changes were observed in the electrode materials even after being cycled for 100 times. Figure 7a,b shows the SEM images of the LFS/C nanocomposite after 100 cycles, which depicts that the particles were unaffected upon lithiation/delithiation. This reveals the appealing structural stability of LFS/C electrodes during cycling. 

Figure 8a,b and the inset shows the TEM images of LFS/C nano composite after 100 cycles. This divulges that the LFS particle was surrounded by the carbon layer, which sustained the crystallite nature and morphology of the material upon cycling.

The XPS spectra for the post-mortem sample are shown in Figure 9a–d. The survey spectrum of the LFS after 100 cycles held similar characteristic peaks as compared to the as-prepared one, except the presence of fluorine (F 1s) peak at 686 eV, which was due to the presence of electrolyte (LiPF_6_) (Figure 9a). As well, the peak of Fe 2p_3/2_ was vanished. It was expected due to the solid electrolyte interface (SEI) formation on the surface of the materials. However, carbon peaks revealed the presence of C=O, C–C and C–Fe respectively at 289, 285 and 283 eV [51].

## 4. Conclusions

The Li_2_FeSiO_4_ cathode material with an orthorhombic structure was successfully synthesized via the polyol method. Evasion of the impure phase of Li_2_SiO_3_ was confirmed via FTIR, Raman and XPS results. TEM demonstrates that the as-prepared cathode material exhibited nano particles about 50 nm in size with spherical morphology with the presence of carbon coating on LFS and the quality of the carbon in term of sp^2^ (ordered graphite) and sp^3^ (disorder graphite) hybridization were identified by using the G and D band, which was observed from the Raman spectrum. Besides, the higher graphitization band enhanced the electronic conductive of LFS. The XPS result proved the divalent state of Fe, therefore the impure phase of Fe was controlled by the polyol solvent. Li_2_FeSiO_4_/C with thin carbon wiring cathode materials achieved 99% coulombic efficiency at 1C with the discharge capacity of 186 at the 1st cycle and 179 mAh·g^−1^ at the 100th cycle. This was achieved only for nanosized uniformly highly symmetry spherical particles of the as-prepared electrode materials with proper utilization of carbon for tailoring the particle with the diffusion coefficient of 1.67 × 10^−16^ cm^2^·s^−1^. This study recommended the suitability of Li_2_FeSiO_4_ cathode for high rate application as evident from the rate performance.

## Figures and Tables

**Figure 1 polymers-11-01703-f001:**
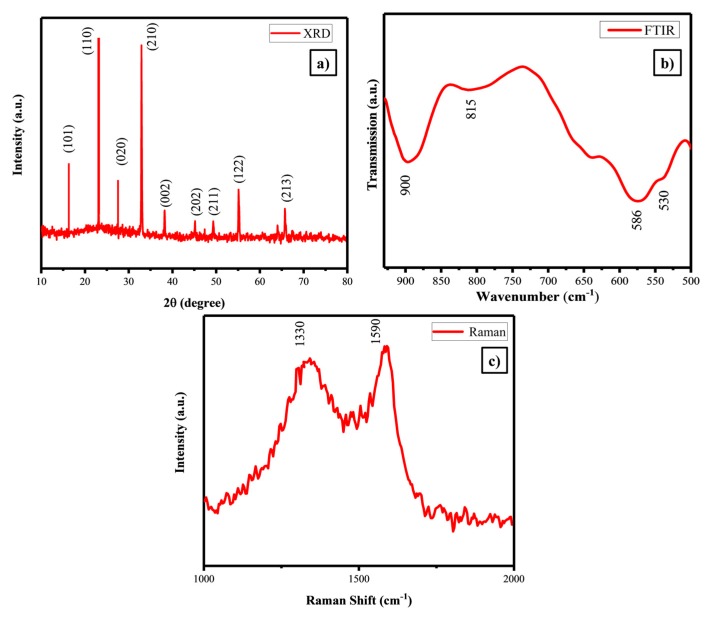
(**a**) XRD pattern, (**b**) FTIR and (**c**) Raman spectrum of the as-prepared Li_2_FeSiO_4_/C.

**Figure 2 polymers-11-01703-f002:**
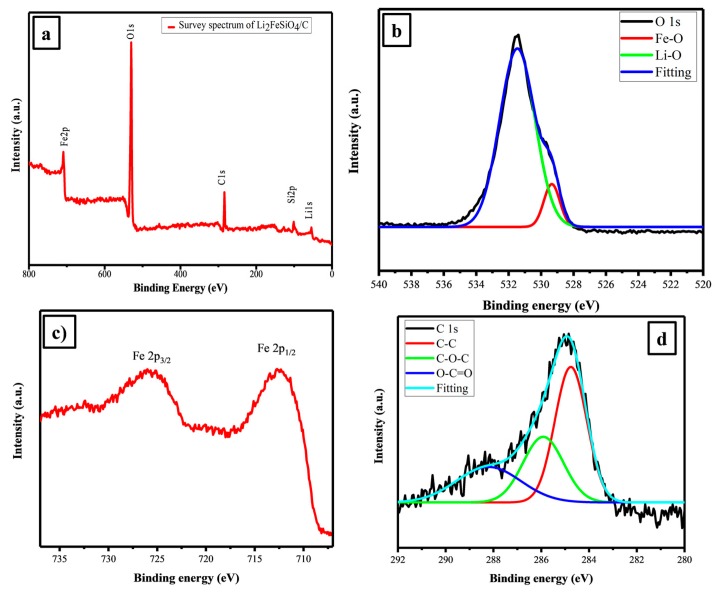
(**a**) Survey, (**b**) O 1s, (**c**) Fe2p and (**d**) C1s X-ray photoelectron spectroscopy (XPS) spectra of Li_2_FeSiO_4_/C.

**Figure 3 polymers-11-01703-f003:**
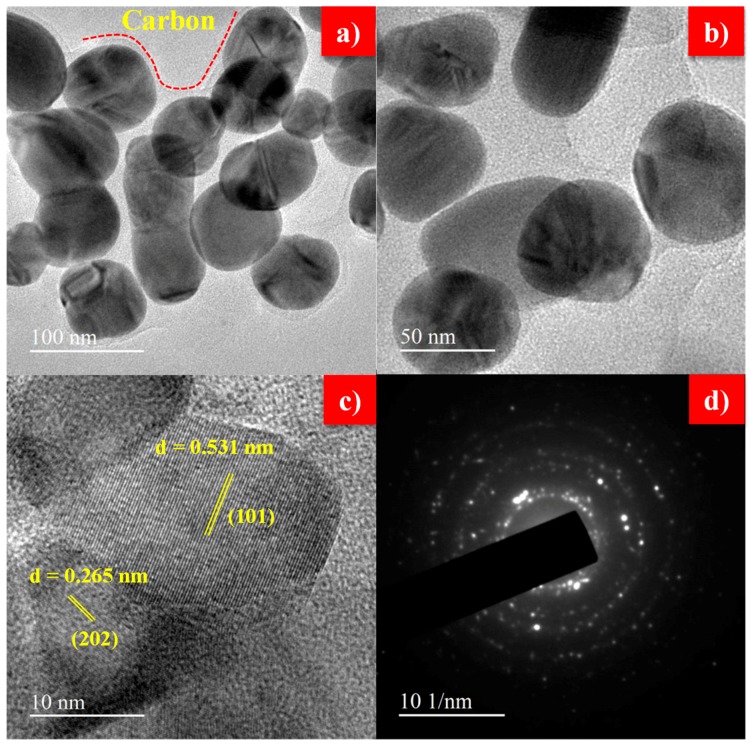
(**a**,**b**) TEM images, (**c**) lattice fringes and (**d**) SAED of as-synthesized Li_2_FeSiO_4_/C.

**Figure 4 polymers-11-01703-f004:**
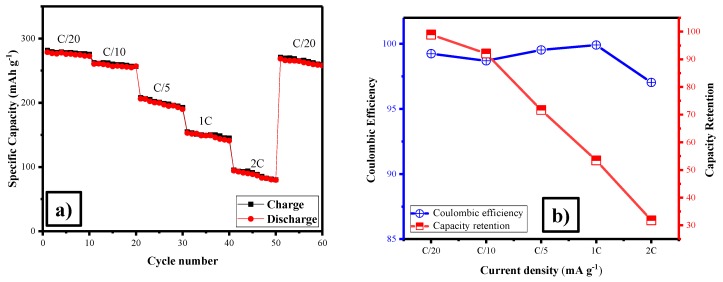
(**a**) Rate performance, and (**b**) columbic efficiency and capacity retention of Li_2_FeSiO_4_/C.

**Figure 5 polymers-11-01703-f005:**
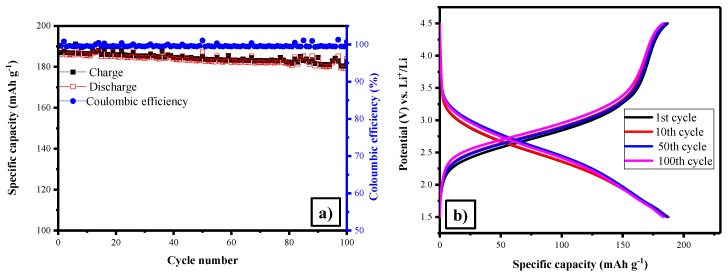
(**a**) Cyclic performance and (**b**) galvanostatic charge discharge profile of Li_2_FeSiO_4_/C.

**Figure 6 polymers-11-01703-f006:**
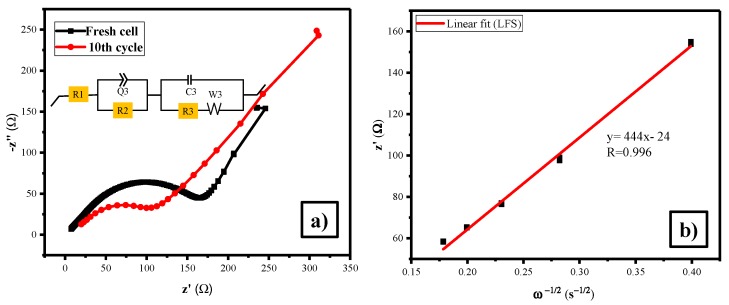
(**a**) Electrochemical impedance spectrum of Li_2_FeSiO_4_/C at fresh and 10th cycle (**b**) lithium diffusion coefficient for LFS.

**Figure 7 polymers-11-01703-f007:**
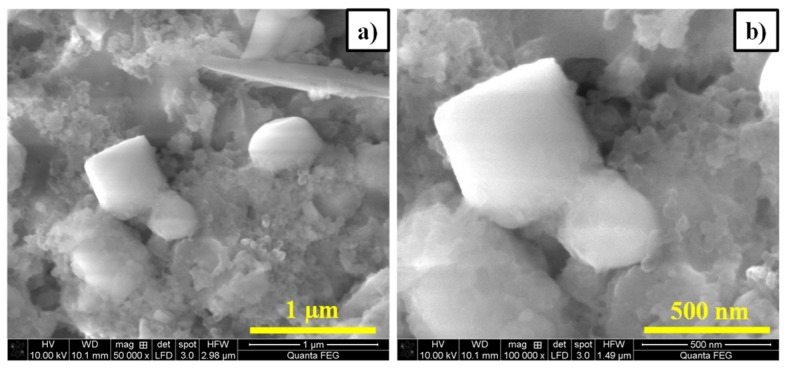
(**a**,**b**) SEM image of Li_2_FeSiO_4_/C after the 100th cycle (post-mortem).

**Figure 8 polymers-11-01703-f008:**
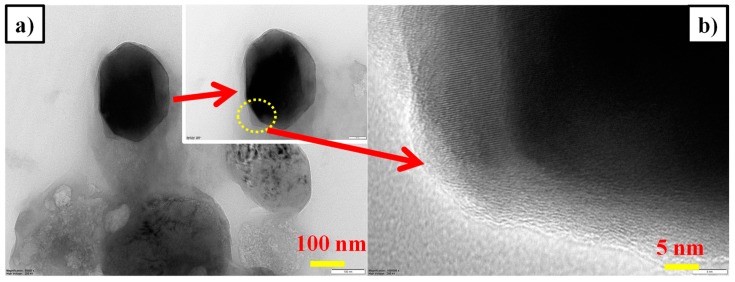
(**a**,**b**) TEM image of Li_2_FeSiO_4_/C after the 100th cycle (post-mortem).

**Figure 9 polymers-11-01703-f009:**
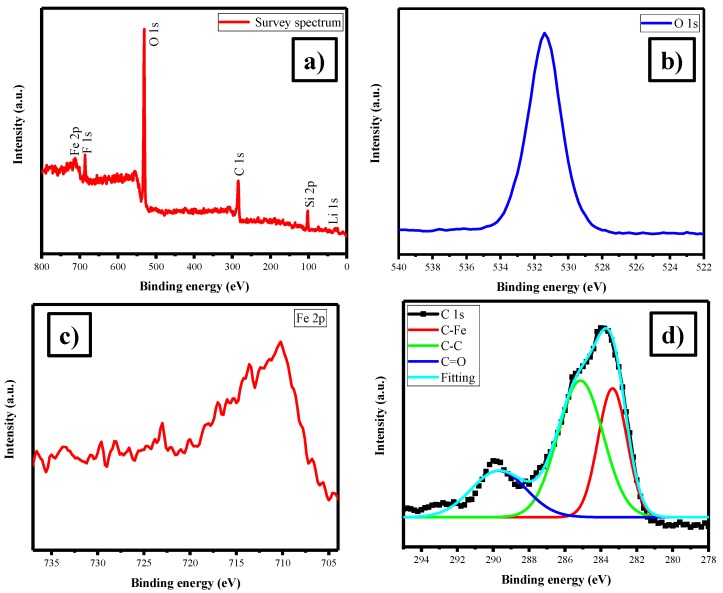
(**a**) Survey, (**b**) O 1s, (**c**) Fe 2p and (**d**) C 1s XPS spectra of Li_2_FeSiO_4_/C after the 100th cycle (post-mortem).

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
