# Peer review of "Carbon Loaded Nano-Designed Spherically High Symmetric Lithium Iron Orthosilicate Cathode Materials for Lithium Secondary Batteries"

_polymers, 2019, doi:10.3390/polym11101703_

Round 1
Reviewer 1 Report
The quality of the paper has been improved sufficiently, and now the paper fulfills requirements for publication. However, there are some minor errors. Please revise the paper carefully again.
For example:
Figure 1: A label of the panel B should be corrected. (LFP=>FTIR)
L122-123: The peaks at 750 and 850 cm-1 is absent in the Figure 1b, if the figure is reported in a transmittance mode.
L125: The numbers of the Raman shifts are different between the sentence and the figure.
L178: mistyped
Author Response
S. No. |
Question |
Response |
1 |
Figure 1: A label of the panel B should be corrected. (LFP=>FTIR) |
Label of the figure 1 B was corrected as mentioned by the reviewer. |
2 |
L122-123: The peaks at 750 and 850 cm-1 is absent in the Figure 1b, if the figure is reported in a transmittance mode. |
As mentioned by the reviewer the error has been rectified as given below. The peak appeared at 815 cm-1 is assigned to the presence of SiO4. |
3 |
L125: The numbers of the Raman shifts are different between the sentence and the figure. |
According to the reviewer’s suggestion, the mistake has been corrected in the revised manuscript as follows. Fig.1c. shows that there are two peaks appeared at 1330 and 1590 cm-1, which are attributed to (D-band sp3–type) and (G-band sp2 type) respectively. |
4 |
L178: mistyped |
Typographic error was corrected. |

Reviewer 2 Report
This manuscript reports a Carbon Loaded Nano-designed Spherically High Symmetric Lithium Iron Orthosilicate Cathode material. The conclusions are sound and the paper is well written. It has potential impact on the battery community. The reviewer thinks that minor revision is suitable.
Paper needs to be corrected by Native English speaker.2. Authors should study any change in morphology of the particles using SEM/TEM after cycling and compare with initial stage.
3. The reviewer would also recommend investigating any change in chemical composition of the particles after cycling using XPS or related techniques.
4. The capacity keeps decaying at higher C rates in Fig 4a. How can this be improved?
5. The following relevant papers should be cited: Adv. Energy Mater. 2017, 7, 1602528, Chem 2016, 1, 287-297, Adv. Energy Mater. 2016, 6, 1600154, Nano Lett. 2016, 16, 1497-1501, Nat. Energy 2016, 1, 15008, Nano Energy 2015, 11, 579-586, Proc. Natl. Acad. Sci. USA 2017, 114, 840-845, Chem. Soc. Rev. 2016, 45, 5605-5634.
Author Response
S. No. |
Question |
Response |
1 |
Paper needs to be corrected by Native English speaker |
As per the suggestion, the linguistic correction has been carried out. |
2 |
Authors should study any change in morphology of the particles using SEM/TEM after cycling and compare with initial stage. |
According to the reviewer’s suggestion, SEM and TEM analyses were performed for the electrodes after cycling. The appropriate images of the electrodes after cycling were presented in the revised manuscripts as Fig.7 and Fig.8. Also, appropriate discussions for the mentioned figures were also added in the revised script. It is noticed from Fig.7 (a-b) that there are no much apparent changes have been observed in the electrode materials even after cycled for 100 times. Fig. 7 (a) and (b) show the SEM images of the LFS/C nanocomposite after 100 cycles, which depicts that the particles are unaffected upon lithiation / delithiation. This reveals the appealing structural stability of LFS/C electrodes during cycling.
Fig. 8 (a, b) and inset show the TEM images of LFS/C nano composite after 100 cycles. This divulges the LFS particle has been surrounded by the carbon layer, which sustains the crystallite nature and morphology of the material upon cycling. |
3 |
The reviewer would also recommend investigating any change in chemical composition of the particles after cycling using XPS or related techniques. |
As per reviewer comment, XPS analysis was performed for the cycled electrode and the results are discussed in the revised manuscript. The XPS spectra for post-mortem sample are shown in Fig.9 (a-d). The survey spectrum of the LFS after 100 cycles holds similar characteristic peaks as compared to the as-prepared one, except the presence of fluorine (F 1s) peak at 686 eV, which is due to the presence of electrolyte (LiPF6) (Fig.9(a)). As well, the peak of Fe 2p3/2 is vanished. It is expected due to the SEI formation on the surface of the materials. However, carbon peaks reveal the presence of C=O, C-C and C-Fe respectively at 289, 285, and 283 eV [51]. |
4 |
The capacity keeps decaying at higher C rates in Fig 4a. How can this be improved? |
It is the inherent property of the materials. The fast kinetics which reduces the diffusion of Li-ion and only makes capacity type effect (i.e., facial adsorption). This is the major factor for slight decay in capacity. It may be expected that this issue can be resolved by means of adding further carbon additives, through structural modification, particle size reduction, etc. |
|
The following relevant papers should be cited: Adv. Energy Mater. 2017, 7, 1602528, Chem2016, 1, 287-297, Adv. Energy Mater. 2016, 6, 1600154, Nano Lett. 2016, 16, 1497-1501, Nat. Energy 2016, 1, 15008, Nano Energy 2015, 11, 579-586, Proc. Natl. Acad. Sci. USA 2017, 114, 840-845, Chem. Soc. Rev. 2016, 45, 5605-5634. |
As per the suggestion made by the referee, all the references are included appropriately in the revised manuscript. 15. Fan, Y.; Yang, Z.; Hua, W.; Liu, D.; Tao, T.; Rahman, M.M.; Lei, W.; Huang, S.; Chen, Y. Lithium-Sulfur Batteries: Functionalized Boron Nitride Nanosheets/Graphene Interlayer for Fast and Long-Life Lithium–Sulfur Batteries (Adv. Energy Mater. 13/2017). Advanced Energy Materials 2017, 7, doi:10.1002/aenm.201770066. 24. Sun, Y.; Zheng, G.; Seh, Zhi W.; Liu, N.; Wang, S.; Sun, J.; Lee, Hye R.; Cui, Y. Graphite-Encapsulated Li-Metal Hybrid Anodes for High-Capacity Li Batteries. Chem 2016, 1, 287-297, doi:https://doi.org/10.1016/j.chempr.2016.07.009. 1. Sun, Y.; Lee, H.-W.; Seh, Z.W.; Zheng, G.; Sun, J.; Li, Y.; Cui, Y. Lithium Sulfide/Metal Nanocomposite as a High-Capacity Cathode Prelithiation Material. Advanced Energy Materials 2016, 6, doi:10.1002/aenm.201600982. 2. Sun, Y.; Lee, H.-W.; Zheng, G.; Seh, Z.W.; Sun, J.; Li, Y.; Cui, Y. In Situ Chemical Synthesis of Lithium Fluoride/Metal Nanocomposite for High Capacity Prelithiation of Cathodes. Nano Letters 2016, 16, 1497-1501, doi:10.1021/acs.nanolett.5b05228. 3. Sun, Y.; Lee, H.-W.; Seh, Z.W.; Liu, N.; Sun, J.; Li, Y.; Cui, Y. High-capacity battery cathode prelithiation to offset initial lithium loss. Nature Energy 2016, 1, 15008, doi:10.1038/nenergy.2015.8. 4. Sun, Y.; Seh, Z.W.; Li, W.; Yao, H.; Zheng, G.; Cui, Y. In-operando optical imaging of temporal and spatial distribution of polysulfides in lithium-sulfur batteries. Nano Energy 2015, 11, 579-586, doi:https://doi.org/10.1016/j.nanoen.2014.11.001. 5. Zhou, G.; Tian, H.; Jin, Y.; Tao, X.; Liu, B.; Zhang, R.; Seh, Z.W.; Zhuo, D.; Liu, Y.; Sun, J., et al. Catalytic oxidation of Li<sub>2</sub>S on the surface of metal sulfides for Li−S batteries. Proceedings of the National Academy of Sciences 2017, 114, 840-845, doi:10.1073/pnas.1615837114. 6. Seh, Z.W.; Sun, Y.; Zhang, Q.; Cui, Y. Designing high-energy lithium–sulfur batteries. Chemical Society Reviews 2016, 45, 5605-5634, doi:10.1039/C5CS00410A. |

Reviewer 3 Report
Thank you for giving me this opportunity to review this manuscript entitled “Carbon Loaded Nano-designed Spherically High Symmetric Lithium Iron Orthosilicate Cathode Materials for Lithium Secondary Batteries” by K. Diwakar et al. They have provided the detailed investigations for Lithium Iron Orthosilicate cathodes for LIBs, the theme of the work is good but lacks in technical point of view. The provided results are highly enthusiastic for materials scientists and LIB industries. However, some of the modifications as a minor revision need to be carried out for the acceptance of this manuscript in Polymers.
The introduction section is very vague. It needs to be modified, should be crisp and deliver the salient features of the orthosilicates based electrodes. Further, to strengthen the introduction part, it is suggested that many previous electrochemical energy storage device would be take in to consideration for comparison for adding into introduction with appropriate references(1166/mat.2016.1337;10.1016/j.memsci.2016.05.010;10.1021/acsenergylett.7b00452;10.1038/s41598-017-11614-1;10.1021/acssuschemeng.8b00090; 10.1007/s10008-016-3466-2; 10.1016/j.cej.2018.10.059)
The author should provide the important electrode parameters, such as loading amount (mg/cm2) and density (g/cm3), which are missing in the main manuscript. Please indicate the loading amount and density information in the updated manuscript
As far as concerning impedance analyses, the author should include its corresponding equivalent circuit for figure 6(a)
The capacity shows already signs of fading after 40 cycles. Compared to other systems, is there any information about long-term stability? Also, there should be a direct comparison against the present electrode system with other orthosilicate system and a standard electrode.
Author Response
S. No. |
Question |
Response |
1 |
The introduction section is very vague. It needs to be modified, should be crisp and deliver the salient features of the orthosilicates based electrodes. Further, to strengthen the introduction part, it is suggested that many previous electrochemical energy storage device would be take in to consideration for comparison for adding into introduction with appropriate references(1166/mat.2016.1337;10.1016/j.memsci.2016.05.010;10.1021/acsenergylett.7b00452;10.1038/s41598-017-11614-1;10.1021/acssuschemeng.8b00090; 10.1007/s10008-016-3466-2; 10.1016/j.cej.2018.10.059) |
We thank the reviewer for the nice suggestion, and the introduction has been modified and enlightened by citing the references in the appropriate places in the revised manuscript. 22. Karuppasamy, K.; A, N.; Karthickprabhu, S.; Hirankumar, G.; Shajan, S. A Brief Review on Integrated (Layered and Spinel) and Olivine Nanostructured Cathode Materials for Lithium Ion Battery Applications. MATERIALS FOCUS 2016, 5, 324-334, doi:10.1166/mat.2016.1337. 20. Karuppasamy, K.; Reddy, P.A.; Srinivas, G.; Tewari, A.; Sharma, R.; Shajan, X.S.; Gupta, D. Electrochemical and cycling performances of novel nonafluorobutanesulfonate (nonaflate) ionic liquid based ternary gel polymer electrolyte membranes for rechargeable lithium ion batteries. Journal of Membrane Science 2016, 514, 350-357, doi:https://doi.org/10.1016/j.memsci.2016.05.010. 40. Ni, J.; Jiang, Y.; Bi, X.; Li, L.; Lu, J. Lithium Iron Orthosilicate Cathode: Progress and Perspectives. ACS Energy Letters 2017, 2, 1771-1781, doi:10.1021/acsenergylett.7b00452. 27. Karuppasamy, K.; Kim, H.-S.; Kim, D.; Vikraman, D.; Prasanna, K.; Kathalingam, A.; Sharma, R.; Rhee, H.W. An enhanced electrochemical and cycling properties of novel boronic Ionic liquid based ternary gel polymer electrolytes for rechargeable Li/LiCoO2 cells. Scientific Reports 2017, 7, 11103, doi:10.1038/s41598-017-11614-1. 41. Wei, H.; Lu, X.; Chiu, H.-C.; Wei, B.; Gauvin, R.; Arthur, Z.; Emond, V.; Jiang, D.-T.; Zaghib, K.; Demopoulos, G.P. Ethylenediamine-Enabled Sustainable Synthesis of Mesoporous Nanostructured Li2FeIISiO4 Particles from Fe(III) Aqueous Solution for Li-Ion Battery Application. ACS Sustainable Chemistry & Engineering 2018, 6, 7458-7467, doi:10.1021/acssuschemeng.8b00090. 25. Karuppasamy, K.; Reddy, P.A.; Srinivas, G.; Sharma, R.; Tewari, A.; Kumar, G.H.; Gupta, D. An efficient way to achieve high ionic conductivity and electrochemical stability of safer nonaflate anion-based ionic liquid gel polymer electrolytes (ILGPEs) for rechargeable lithium ion batteries. Journal of Solid State Electrochemistry 2017, 21, 1145-1155, doi:10.1007/s10008-016-3466-2. 30. Tian, M.; Hu, L.; Huang, Z.; Li, M.; Yang, J.; Wang, Z.; Li, J.; Lin, X.; Mu, S. Tri-phase (1-x-y) Li2FeSiO4·xLiFeBO3·yLiFePO4 nested nanostructure with enhanced Li-storage properties. Chemical Engineering Journal 2019, 358, 786-793, doi:https://doi.org/10.1016/j.cej.2018.10.059. |
2 |
The author should provide the important electrode parameters, such as loading amount (mg/cm2) and density (g/cm3), which are missing in the main manuscript. Please indicate the loading amount and density information in the updated manuscript |
As indicated by the referee, the following information has been included in the revised script. 3.2 mg cm-2 of electrode materials was loaded for performing the electrochemical characterization. |
3 |
As far as concerning impedance analyses, the author should include its corresponding equivalent circuit for figure 6(a) |
Equivalent circuit was inserted in the EIS (Fig. 6a) as indicated by the reviewer. |
4 |
The capacity shows already signs of fading after 40 cycles. Compared to other systems, is there any information about long-term stability? Also, there should be a direct comparison against the present electrode system with other orthosilicate system and a standard electrode. |
Direct comparison against the present electrode system with other orthosilicate system has been referred. The capacity retention of 96% was attained for this material, which is comparatively higher than reports for other orthosilicates [54,55]. |
